# Previously undetected super-spreading of *Mycobacterium tuberculosis* revealed by deep sequencing

Robyn S Lee[1,2,3]*, Jean-François Proulx[4], Fiona McIntosh[5], Marcel A Behr[5], William P Hanage[2,3]

[1]Epidemiology Division, Dalla Lana School of Public Health, University of Toronto, Toronto, Canada; [2]Center for Communicable Disease Dynamics, Harvard TH Chan School of Public Health, Boston, United States; [3]Department of Epidemiology, Harvard TH Chan School of Public Health, Boston, United States; [4]Nunavik Regional Board of Health and Social Services, Kuujjuaq, Canada; [5]The Research Institute of McGill University Health Centre, Montréal, Canada

**Abstract** Tuberculosis disproportionately affects the Canadian Inuit. To address this, it is imperative we understand transmission dynamics in this population. We investigate whether 'deep' sequencing can provide additional resolution compared to standard sequencing, using a well-characterized outbreak from the Arctic (2011–2012, 50 cases). Samples were sequenced to ~500–1000x and reads were aligned to a novel local reference genome generated with PacBio SMRT sequencing. Consensus and heterogeneous variants were identified and compared across genomes. In contrast with previous genomic analyses using ~50x depth, deep sequencing allowed us to identify a novel super-spreader who likely transmitted to up to 17 other cases during the outbreak (35% of the remaining cases that year). It is increasingly evident that within-host diversity should be incorporated into transmission analyses; deep sequencing may facilitate more accurate detection of super-spreaders and transmission clusters. This has implications not only for TB, but all genomic studies of transmission - regardless of pathogen.

*For correspondence:
robyn.s.c.lee@gmail.com

Competing interests: The authors declare that no competing interests exist.

## Introduction

Tuberculosis (TB) in Canada is highest among the Inuit, an Indigenous population with a rate over 300 times that of the non-Indigenous Canadian-born population in 2016 (*Inuit Tapiriit Kanatami, 2018*). Canada recently set a goal of TB elimination in the Inuit by 2030, (*Inuit Tapiriit Kanatami, 2018*) which will not be achieved without halting ongoing transmission. Previous studies have used genomic data either alone or in conjunction with classical epidemiology to investigate TB transmission dynamics in the Canadian North, (*Tyler et al., 2017*; *Lee et al., 2015a*; *Lee et al., 2015b*) with the aim of identifying clusters to help guide public health interventions. Thus far, such studies have relied on identifying consensus single nucleotide polymorphisms (cSNPs), consistent with prevailing methodology in this field.

Recent studies suggest that incorporation of within-host diversity into genomic analyses may provide greater resolution of transmission than cSNP-based approaches alone (*Worby et al., 2017*; *Martin et al., 2018*; *Meehan et al., 2019*; *Séraphin et al., 2019*). This may be particularly important for investigation of outbreaks occurring over short time scales and/or in settings such as the Canadian North, where the genetic diversity of circulating strains is especially low. In both of these circumstances, it is common to find many samples separated by zero cSNPs, hindering accurate source ascertainment. To investigate this hypothesis, we used deep sequencing (i.e., to ~10-20 fold more

**eLife digest** In Canada, tuberculosis disproportionately affects the Inuit, a group of indigenous people inhabiting the Arctic regions. Canada is aiming to eliminate tuberculosis among the Inuit by 2030. One way to help stop transmission and prevent future outbreaks is to trace how and where the disease spreads using DNA sequencing. This information can then be used by public health organizations to identify possible interventions.

Typically, the DNA of the bacterium that causes tuberculosis – *Mycobacterium tuberculosis*, or Mtb for short – is sequenced 50–100 times and a consensus DNA sequence is then generated for each patient from this data. These consensus DNA sequences are then compared to help piece together who infected whom. Recently, scientists have realized that the bacteria a person is infected with may have different DNA sequences due to people being infected with more than one bacterium or the bacterium developing variations in its genome after the infection. However, current DNA sequencing practices may miss these differences, making it harder to trace how the disease spreads.

Now, Lee et al. show that sequencing the DNA of Mtb from an infected person 500–1000 times (i.e. ~10-20 times more than usual) makes it easier to detect genetic differences and determine how tuberculosis spreads. This approach, also known as 'deep sequencing', was used to analyze DNA samples of Mtb collected from about 50 people during an outbreak of tuberculosis in 2011-2012, which had previously undergone standard DNA sequencing.

This deep sequencing approach identified a 'super-spreading event' where one person had likely transmitted tuberculosis to up to 17 others during the outbreak. Lee et al. found that most of these people had visited the same 'gathering houses' which are social venues in the community. Implementing targeted public health interventions at these sites may help stop future outbreaks.

To fully understand how useful this method will be for tracking the spread of tuberculosis, deep and routine sequencing will need to be compared against each other in different settings and outbreaks. Furthermore, the approach used in this study may be useful for tracking the transmission of other infectious diseases.

than standard, or 500-1000x) to re-evaluate transmission in a densely-sampled outbreak in Nunavik, Québec.

This outbreak, which has been previously described, (*Lee et al., 2015b*; *Lee et al., 2016*) comprised 50 microbiologically-confirmed cases of TB who were diagnosed in a single Inuit community between 2011–2012 - a rate of 5,359/100,000 for that year. Genomic epidemiology analyses using sequencing depths of ~50x that are standard in such work, identified multiple clusters of transmission in this outbreak, (*Lee et al., 2015b*) however, there was insufficient genetic variation detected to infer precise person-to-person transmission events within these subgroups, given the short time frame and low mutation rate of *M. tuberculosis* (~0.2–0.3 SNPs/genome/year for Lineage 4; *Menardo et al., 2019*). In this study, we illustrate how within-host diversity can be incorporated into transmission analyses. In doing so, we find new features of the transmission networks in this community, in particular, identifying a previously unrecognized super-spreading event. We highlight a potential role for deep sequencing in public health investigations, with implications for TB control in Canada's North as well as other high-transmission environments.

## Materials and methods

### Study subjects

All 50 samples from the 2011–2012 outbreak (*Lee et al., 2015b*) were eligible for inclusion, as well as samples from all cases (n = 15) diagnosed in same village in the preceding five years (2007 onwards), 13/15 of which were caused by the same strain of *M. tuberculosis* (the 'Major [Mj]-III' sublineage; *Lee et al., 2015a*). There were two episodes of recurrent TB (i.e., where an individual had microbiologically-confirmed TB once, was cured, but developed TB again during the study period); otherwise, all samples are from unique individuals. All cases had pulmonary TB that was Lineage 4

(Euro-American; *Lee et al., 2015b*). Cross-contamination was ruled out as described in *Lee et al. (2015b)*.

## DNA extraction and sequencing

Samples were cultured once on Middlebrook 7H10 agar and plate sweeps were collected for DNA extraction using the van Soolingen method (*van Soolingen et al., 1991*). Genomic DNA was quantified using the Quant-iT PicoGreen dsDNA Assay (ThermoFisher Scientific, Massachusetts, USA). Library preparation and sequencing were done at the McGill University/Genome Québec Innovation Centre. The Illumina HiSeq 4000 was used to produce paired-end 100 bp reads. To obtain the depth of coverage needed for this study (~500–1000x for deep sequencing, compared to ~50–100x as routinely done by public health), pooled libraries were run on four independent lanes.

## Bioinformatics

FastQC (v.0.11.5, https://www.bioinformatics.babraham.ac.uk/projects/fastqc/) was used to assess sequencing data quality and reads were trimmed to remove low-quality bases using Trimmomatic (v.0.36 *Bolger et al., 2014*). Kraken (v.1.1 *Wood and Salzberg, 2014*) was then used to identify potential contamination with the miniKraken database (minikraken_20171019). Reads classified as '*Mycobacterium tuberculosis* complex' (MTBC) were extracted using Seqtk (v.1.2, Li H, available at: https://github.com/lh3/seqtk).

Reads were then aligned using the Burrows Wheeler Aligner MEM algorithm (v.0.7.15 *Li, 2013*) to the H37Rv reference (NC_000962.3 in the National Center for Biotechnology Information [NCBI] RefSeq database) and sorted using Samtools (v.1.5 *Li et al., 2009*). Analyses were later repeated using a local reference genome (described below). Reads were with ambiguous mappings were excluded, as were reads with excessive soft-clipping (i.e., more than 20% of read length) based on our previous work (*Martin et al., 2018*). Duplicate reads were marked using Picard MarkDuplicates (v.2.9.0, https://broadinstitute.github.io/picard/) and reads were locally re-aligned around indels using Genome Analysis ToolKit (GATK, v.3.8 *McKenna et al., 2010*). All sites were called using GATK's Unified Genotyper algorithm, with the -d 1500 to avoid downsampling to 250 (done by default with this tool during variant calling). Variants (cSNPs and heterogenous SNPs [hSNPs]) versus H37Rv were annotated using snpEff (v.4.3t *Cingolani et al., 2012*).

Variants were filtered for quality using custom Python scripts (v.3.6) with the following thresholds: Phred < 50, Root Mean Squared Mapping Quality (RMS-MQ) $\leq$ 30, depth (DP) < 20, Fisher Strand Bias (FS) $\geq$ 60 and read position strand bias (ReadPos) < $-8$ (*Martin et al., 2018*). cSNPs were classified as positions where $\geq$ 95% of reads were the alternative allele (ALT), hSNPs were classified as positions where > 5% and < 95% of reads were ALT, and positions with the ALT present in $\leq$5% of reads were classified as 'reference'. We also compared inferences of transmission from this analysis to i) when these thresholds were increased to the minimum values among cSNPs in the initial H37Rv analysis, and ii) when cSNPs were classified using a threshold of $\geq$ 99%, and hSNPs were classified when 1% < ALT < 99%, in order to assess the robustness of inferences to different filtering protocols.

Low-quality variants, variants in proline-proline-glutamic acid (PE) and proline-glutamic-acid/polymorphic-guanine-cytosine-rich sequence (PE_PGRS) genes, transposons, phage and integrase, and positions with missing data, were excluded. All samples were drug-susceptible, except for MT-6429, which was rendered resistant to isoniazid by a frameshift deletion at position 1284 in the catalase-peroxidase gene *katG*. As such, positions associated with drug resistance were not masked in this analysis. Alignments with informative hSNPs were reviewed using Tablet (v.1.17.08.17, *Milne et al., 2013*).

Concatenated core cSNP alignments were made using snp-sites -c (v.2.4.0 *Page et al., 2016*), with positions with hSNPs excluded. Pairwise cSNP distances between samples were computed using snp-dists (v.0.6, available at https://github.com/tseemann/snp-dists). The frequency of hSNPs at each position in the genome was tabulated and hSNPs were reviewed to identify variants shared between samples.

## Phylogenetics and clustering

Core cSNP alignments were used to generate maximum likelihood trees using IQ-Tree (v.1.6.8 *Nguyen et al., 2015*). Model selection was based on the lowest Bayesian Information Criterion. Hierarchical Bayesian Analysis of Population Structure (*Cheng et al., 2013*) was run in R (v.3.5.2) to identify clusters. Phylogenetic trees were visualized using Interactive Tree of Life (*Letunic and Bork, 2016*).

## Single molecule Real-Time (SMRT) sequencing and assembly

To examine the influence of potential alignment errors in identification of hSNPs, we used SMRT sequencing with the PacBio RSII platform to create a local reference genome for the outbreak. Sample MT-0080 was chosen for sequencing because this was previously identified as the probable source for as many as 19 of the 50 cases diagnosed in 2011–2012 (*Lee et al., 2015b*). Prior to sequencing, the culture was grown on a Middlebrook 7H10 agar plate. A single colony was then selected and grown further in 3 mL of Middlebrook 7H9 Broth to provide sufficient DNA for SMRT sequencing and Illumina MiSeq (for polishing of the long-read assembly). DNA for SMRT sequencing was extracted using the MagAttract High Molecular Weight DNA Kit from Qiagen (Maryland, USA). High molecular weight fragments were verified using gel electrophoresis. Library preparation and sequencing were then done at the McGill University/Genome Québec Innovation Centre. Prior to sequencing, fragment size was evaluated using a BioAnalyzer and the BluePippin system (Sage Science, Massachusetts, USA) was used for size selection. DNA for Illumina MiSeq was extracted using the van Soolingen method, as previous (*van Soolingen et al., 1991*).

Long-reads were assembled and corrected using Canu (v.1.7.1 *Koren et al., 2017*). Pilon (v.1.23 *Walker et al., 2014*) was then used to polish the assembly using the Illumina MiSeq reads from the same colony. This was re-run until no further corrections were possible. Quast (v.5.0.2, *Gurevich et al., 2013*) was used to evaluate assembly quality. RASTtk (v.2.0 *Brettin et al., 2015*) was used for annotation, to identify regions for masking as previous.

## Epidemiological data

Epidemiological and clinical data were collected on all cases and contacts using standardized questionnaires as part of the routine public health response, described previously in *Lee et al. (2015b)*; *Lee et al. (2016)*.

## Statistical analyses

A two-sample test of proportions was used to compare overall proportions across references, and the Wilcoxon Signed Rank test was used to compare paired SNP distances. Analyses were done in Stata (v.15, StataCorp, College Station, TX, USA).

# Results

62/65 (95·4%) available TB samples from cases diagnosed between 2007–2012 were successfully sequenced and passed quality control. This included 48/49 (98·0%) of the samples with an identical Mycobacterial Interspersed Repetitive Units Variable Number Tandem Repeats (MIRU-VNTR) pattern during the outbreak year. The remaining three samples could not be re-grown. Reads that were non-MTBC were removed (*Source data 1*) and there was no obvious association between percent contamination and hSNP frequency. Epidemiological and clinical data on all outbreak cases are described in *Lee et al. (2015b)*.

Average genome coverage and depth across the H37Rv reference was 98·64% [SD 0·07%] and 714·53 [SD 92·68], respectively. Our primary filtering protocol yielded 51,430 cSNPs and 4,897 hSNPs across all individual samples (*Source data 2*). Excluding positions that were invariant compared to the reference or where any sample was missing and/or was low-quality resulted in a core alignment of 860 cSNP positions and 136 hSNP positions (note, these are not mutually exclusive, as positions with cSNPs in some samples may have hSNPs in others).

42 positions had hSNPs that were shared across all 62 samples (*Table 1*, *Source data 3*). Depth of coverage at these positions was, on average, 39% higher than the average depth across the same sample (SD 36·7%, *Source data 4*). Along with manual review of alignments (*Figure 1*), this

**Table 1.** Comparison of alignments to H37Rv and MT-0080_PB.

Based on these filters: Phred < 50, Root Mean Square Mapping Quality (RMS-MQ) ≤ 30, depth (DP) < 20, Fisher Strand Bias (FS) ≥ 60 and read position strand bias (ReadPos) < −8 and an allelic fraction of ≥ 95% for cSNPs, with hSNPs classified when 5% < ALT < 95%. Quality metrics for the individual cSNPs/hSNPs identified in each sample are given in *Source data 2*.

| | H37Rv (4,411,532 bp) | MT-0080_PB (4,426,525 bp) | P value |
|---|---|---|---|
| Number of positions according to reference genome | | | |
| Invariant reference across all samples, n (%) | 4,018,786 (91·10%) | 4,084,195 (92·27%) | <0·00005 [a] |
| Position was missing/low quality in at least one sample, n (%) | 391,761 (8·88%) | 342,179 (7·73%) | <0·00005 [a] |
| Position was an c/hSNP in at least one sample, n (%) | 985 (0·22%) | 152 (0·00%) | <0·00005 [a] |
| Shared cSNPs across all samples, n (%) | 764 (0·02%) | 1 (0·00%) | <0·00005 [a] |
| Shared hSNPs across all samples, n (%) | 42 (0·00%) | 0 (0%) | <0·00005 [a] |
| Core pairwise distances | | | |
| Core cSNPs vs. reference, median (range) | 791 (790–792) | 3 (1–65) | <0·00005 [b] |
| Core cSNPs between samples, median (range) | 3 (0–64) | 3 (0–66) | <0·00005 [b] |

[a] Two sample test for difference in proportions.
[b] Wilcoxin Signed Rank test.

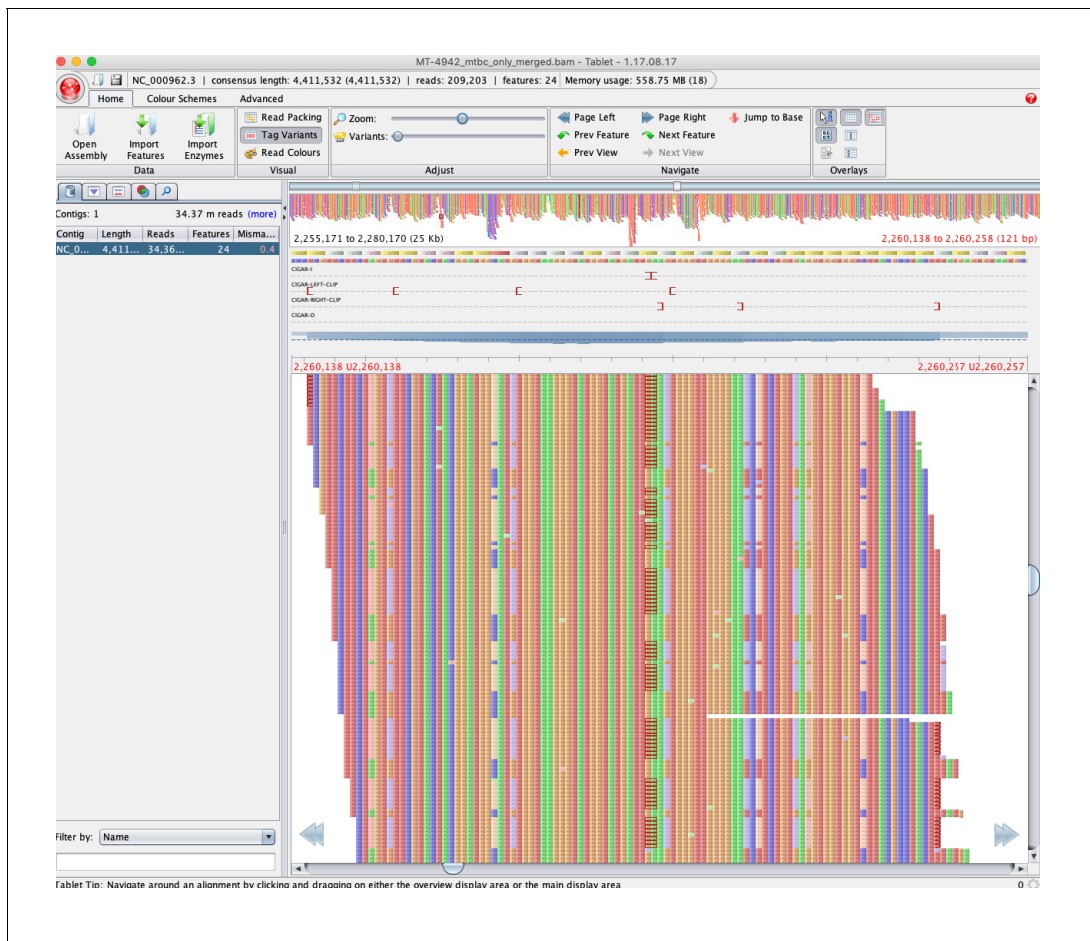

**Figure 1.** Pileup of reads showing hSNPs suspected to be due to alignment error as listed in *Source data 3*, with MT-4942 used as an example and zoomed on position 2,255,171 to 2,280,170 in H37Rv (National Center for Biotechnology Information RefSeq Database Accession NC_000962.3). Binary Alignment Map (BAM) file were loaded into Tablet (v.1.17.08.17, *Milne et al., 2013*) to visualize the pileup compared to H37Rv.

suggested that many of these were false positives, potentially due to alignment error (e.g., from underlying structural variation in our samples compared to the H37Rv reference).

To address this, we generated a local reference genome for the outbreak, MT-0080_PB. Quality metrics for the MT-0080_PB assembly are given in *Source data 5*. Compared to H37Rv, mean genome coverage and depth were higher with MT-0080_PB (at 99·33% [SD 0·09%] and 717·07 [SD 93·01], respectively), fewer positions were missing/low-quality (p < 0·00005, *Table 1*), and overall, fewer variable positions were detected (p < 0·00005). While core cSNP distances were similar between samples regardless of the reference (*Table 1*), the number of hSNPs was greatly reduced using MT-0080_PB (*Source data 2*); while 4,897 hSNPs were identified across all individual samples using H37Rv, only 125 hSNPs were identified using MT-0080_PB. There were also no hSNPs shared across all 62 samples using MT-0080_PB. Together, these findings support our hypothesis that alignment error is responsible for many of the detected variants, and indicate a local reference is important for accurate identification of hSNPs. All further results presented are based on the MT-0080_PB alignment.

A maximum likelihood tree was generated from 90 core cSNP positions (excluding sites invariant across all samples and the reference) compared to MT-0080_PB (*Figure 2*). All 62 samples (historical and outbreak) were included, for comparison with our previous work. (*Lee et al., 2015b*) Consistent with this, (*Lee et al., 2015b*) hierBAPS identified two main sub-lineages ('Mj-V' and 'Mj-III' per *Lee et al., 2015a*), with three sub-clusters (Mj-IIIA/B/C).

## hSNPs identify a novel super-spreading event and more accurately resolve transmission clusters

The core cSNPs and hSNPs between samples are shown in *Source data 6*, with the sub-groups identified in the original analysis indicated. Overlaying hSNPs with the cSNP-based analysis revealed a novel super-spreader (MT-504) in Cluster Mj-IIIB, undetected by genomic epidemiology analyses relying on lower sequencing depth. (*Lee et al., 2015b*).

In brief, our previous analysis using routine sequencing depth had suggested that Mj-IIIB was comprised of two distinct transmission networks (which we refer to as 'subgroups' for consistency with our earlier work); the first subgroup consisted of five cases diagnosed between December 2011 and October 2012, while the second subgroup consisted of 13 cases diagnosed between March and November 2012. Epidemiologic curves for these subgroups are given in *Figure 2—figure supplement 1A*. These two subgroups were distinguished from one another based on the presence or absence of a shared cSNP (at position 276,685 according to H37Rv/276,544 in the MT-0080_PB alignment, *Source datas 6*, *7*, *8*); all samples in the subgroup of five cases shared an alternative 'C' allele at this position, while all samples in the subgroup of 13 cases shared the reference 'A' allele. Given the short time period, low mutation rate of TB, and overall low diversity of strains circulating in the village, we would expect 0 SNPs to accrue in recent transmission, refuting transmission between these subgroups. In the original analysis, MT-504 was identified as the probable source for the subgroup of five cases. This individual was diagnosed in late 2011 with smear-positive, cavitary disease, and had attended the same local community 'gathering houses' (social venues specifically identified by public health during the outbreak) as all four others in this subgroup. For the second subgroup of 13 cases, MT-2474 was identified as the probable source, as this was the first smear-positive case in this subgroup (diagnosed in May 2012, *Figure 2—figure supplement 1A*).

In contrast to the analysis based on routine sequencing data, our in-depth investigation of within-host diversity using deep sequencing data revealed that MT-504 harboured both the alternative allele (563 reads [80·9%]) shared by all members of the subgroup of five as well as the reference allele (133 reads [19·1%]), shared by all members of the subgroup of 13 (*Figure 2—figure supplement 1B*). Given that MT-504 was the first contagious case diagnosed in Mj-IIIB (*Figure 2—figure supplement 1A*), and all 13 cases in the second subgroup had attended or resided in a gathering house (with 9/13 [69·2%] reporting attendance at the same houses as MT-504), this strongly suggests that MT-504 is in fact the most probable source for both subgroups.

## hSNP analysis adds support for suspected transmission

Sample 68995 and MT-5543 were from 2007, and were the only strains from the Mj-VA sub-lineage in this village. Previous analysis indicated Mj-VA strains from other villages were distantly related,

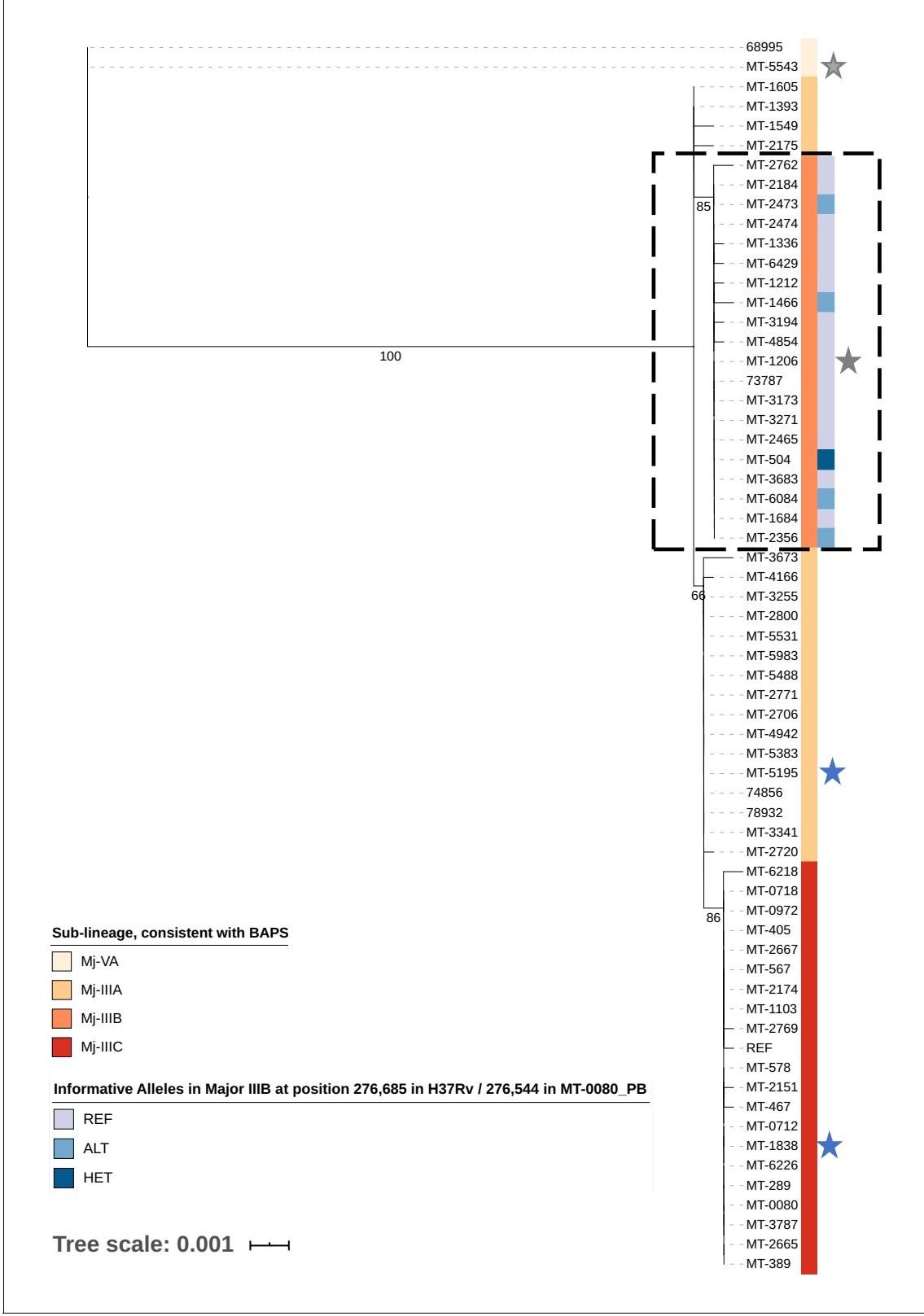

**Figure 2.** Transmission of *M. tuberculosis* in village K. Maximum likelihood tree of 62/65 cases diagnosed between 2007–2012 in village K based on consensus single nucleotide polymorphisms (cSNPs). After aligning to a local reference, MT-0080_PB, cSNPs were identified based on a minimum threshold of ≥95% of reads supporting the alternative allele. A core cSNP alignment was then produced with 90 positions.and IQ-Tree (v.1.6.8 **Nguyen et al., 2015**) was used to generate the tree using a KP3 model with correction for ascertainment bias. Model selection was based on the

*Figure 2 continued on next page*

*Figure 2 continued*

lowest Bayesian Information Criterion. 1000 bootstrap replicates were done; only p values > 60% are shown. Clusters were identified using hierarchical Bayesian Analysis of Population Structure (*Cheng et al., 2013*). These clusters were consistent with the sub-lineages previously identified in *Lee et al. (2015a)*; *Lee et al. (2015b)*, thus only sub-lineage names are indicated (Major sub-lineages [Mj]-IIIA, B, C, and Mj-VA). Only Mj-IIIA/B/C were present in 2011–2012; Mj-IIIA was first seen in village K in 2007, IIIB was first seen in 2009, and IIIC was first seen in 2012. Alleles informative for transmission in Mj-IIIB, identified using deep sequencing, are indicated. Between 2007–2012, there were two individuals who had a second episode of TB; stars are used to highlight these samples, with a different colour for each patient. MT-0080 is included in the alignment as the deep sequencing data from a sweep of all colonies identified a cSNP compared to the MT-0080_PB reference, which itself was generated from a single colony pick.

The online version of this article includes the following figure supplement(s) for figure 2:

**Figure supplement 1.** Comparison of epidemiologic inferences using 'routine' versus 'deep' sequencing.

---

(*Lee et al., 2015b*) while these two samples were separated from one another by zero core cSNPs. This suggests direct transmission between these historical cases, a hypothesis strongly supported by hSNP analysis, as the samples share hSNPs that are not found in any other sample in the dataset. These hSNPs were present even when highly conservative filtering thresholds were used (*Source data 7*), but were not included when using H37Rv as the reference - potentially due to differences in annotation and subsequent filtering.

## Potential utility for discriminating TB recurrence

Six individuals had TB recurrence in 2011–2012. Paired samples were available for two of these (Patient 1: samples MT-5195 [Mj-IIIA] in 2007 and MT-1838 [Mj-IIIC] in 2012; Patient 2: samples MT-5543 [Mj-VA] in 2007 and MT-1206 [Mj-IIIB] in 2012, *Figure 2*). Clinically, both patients had new lesions detected at their second episode, compared to their previous chest x-rays. cSNP-based analyses suggested their second episodes of TB were due to re-infection with a new strain, rather than relapse with the strain causing their original disease. Investigation of within-host diversity strongly supported this conclusion; using deep sequencing, we verified that there was a single, different strain present at both baseline and their second episodes of TB. There was no evidence for mixed infection at either baseline or second episode with these strains, more definitively ruling out relapse in this low diversity setting (*Source datas 6*, *7*, *8*).

## Impact of altering cSNP and hSNP thresholds

To ensure we were not missing lower frequency variants using the prior cSNP/hSNP thresholds, we re-ran our analysis such that hSNPs were classified when 1% < ALT < 99%. Quality scores for individual cSNPs and hSNPs are given in *Source data 9* and the core cSNP/hSNP alignment is shown in *Source data 8*. While our primary analysis using a threshold of ≥95% for cSNPs identified a single cSNP (A to G) shared across all samples compared to MT-0080_PB, close examination of the MT-0080 deep sequencing data (obtained using DNA from a sweep of the plate) showed that this sample had both alleles at this position, with only the minority 'A' allele (33 reads/1189 [2·8%]) isolated for SMRT sequencing. Based on this, we recommend sequencing samples both using a clean sweep (with an alternative sequencing platform) and a single colony pick when generating a reference genome for TB, as using the latter alone may introduce error and affect epidemiological inferences. With this exception, no other informative hSNPs were detected using these thresholds.

## Discussion

As the TB epidemic continues among the Canadian Inuit, targeted public health interventions are essential to halt ongoing transmission. In order to do so, it is important that transmission events and associated risk factors are accurately identified. Our previous work suggested that hSNP analysis could enhance resolution of TB transmission (*Martin et al., 2018*). To investigate how this approach could be applied for TB control, we used deep sequencing to re-examine a major TB outbreak in the Canadian Arctic.

Several recent studies, including work by our group (*Martin et al., 2018*), have shown that *M. tuberculosis* within-host diversity can be transmitted between individuals (*Séraphin et al., 2019*; *Guthrie et al., 2019*). Using deep sequencing data allowed us to better identify this diversity in a Nunavik outbreak compared to previous analyses with standard sequencing depth, (*Lee et al.,*

*2015a*; *Lee et al., 2015b*) and facilitated detection of a novel super-spreading event, where one source case may have transmitted to ~1/3 of the other cases diagnosed between 2011 to 2012. This was in addition to a previously identified super-spreader linked to 19 secondary cases - suggesting up to 75% of the outbreak (36/48, excluding the putative super-spreaders) may be attributable to these events. Super-spreading has been described in a number of pathogens, (*Stein, 2011*) including TB (*Kline et al., 1995*). Our findings suggest this can play an important role in driving TB outbreaks, and that accurate detection of super-spreading events is important for informing appropriate public health interventions. In the case of MT-504, as nearly all of the secondary cases had attended the same local community gathering houses as the putative source, this strongly suggests these venues play an important role in facilitating transmission in this setting.

Several studies have used genomics to investigate TB recurrence, (*Witney et al., 2017*; *Bryant et al., 2013*; *Guerra-Assunção et al., 2015*) however, the methods used to assess for mixed infection at either time point have been inconsistent and may not be sufficient to discriminate recurrence in settings with low strain diversity. In this analysis, we provide proof-of-principle that deep sequencing can potentially help rule out relapse. The distinction between relapse and re-infection is important at individual and population levels; high rates of relapse in a community would indicate a problem with treatment or adherence, potentially warranting changes to clinical management, while re-infection would indicate the need for public health interventions such as active case finding. Also, individuals in Nunavik who have had prior treatment for active TB disease in the past are also not routinely offered prophylaxis on re-exposure, based on historical data suggesting ~80% protection is afforded by prior infection (*Menzies, 1997*). The degree to which re-infection drives recurrence in Nunavik is currently unknown, but if re-infection is the primary cause, this clinical practice may need to be re-evaluated. A population-level genomic epidemiology study is currently underway to evaluate this.

To use deep sequencing to investigate within-host diversity, it is critical we minimize false positive hSNPs. We have shown that using a local strain as a reference can not only reduce error, but improves detection of epidemiologically-informative variants. Genomic differences between outbreak strains and H37Rv have been previously illustrated by *Roetzer et al. (2013)*; *O'Toole and Gautam (2017)*, with *O'Toole and Gautam (2017)* warning that clinical TB strains may be needed to fully detect virulence genes in reference-based analyses. Our analysis suggests these may also be warranted for hSNP analysis; where possible, we suggest using long-read sequencing to generate complete and local reference genomes.

Overall, our study has a number of strengths. Firstly, we had access to a densely-sampled outbreak, which was previously investigated using 'standard' sequencing depth and for which detailed epidemiological data was available. This allowed us to readily compare methodological approaches, showing how and when deep sequencing might be beneficial for public health. In this study, we have also identified important methodological considerations for hSNP detection, with implications for transmission analyses, but also potentially for resistance prediction as well (*Liu et al., 2015*). Finally, the use of long-read data has allowed us to completely assemble a novel TB genome from Nunavik. This will serve as a valuable resource for future studies of transmission in Nunavik (given the low strain diversity in the region *Lee et al., 2015a*), as well as other Inuit territories.

A potential limitation of this work is that, given the historical nature of the outbreak, deep sequencing was done using DNA extracted from culture. Due to methodological challenges of sequencing directly from sputum, (*Brown et al., 2015*; *Votintseva et al., 2017*; *Doyle et al., 2018*) few studies have examined the effect of culture on genome diversity. A recent study by *Shockey et al. (2019)*., which analyzed allelic variation among reads from individual samples, suggests that some diversity may be lost during the culturing process. However, several studies looking at potential transmission (*Votintseva et al., 2017*; *Doyle et al., 2018*; *Nimmo et al., 2019*) found results were congruent between cSNP analyses from culture versus raw samples. In terms of hSNPs, *Votintseva et al. (2017) Doyle et al. (2018)*; *Nimmo et al. (2019)* reported detecting fewer hSNPs with sequencing from culture versus from sputum, in *Nimmo et al. (2019)*, the median hSNPs was only 4.5 versus 5 hSNPs, respectively – a difference that may not be clinically significant, regardless of statistical significance. Given the inconsistency of results and paucity of data, further study is needed to understand how hSNP diversity may be affected by the culturing process, and to assess whether this affects inferences of transmission. We note that it is likely that enhanced detection of the hSNPs present in sputum would improve the resolution over that which we present in this work.

Another potential limitation is that, while we can compare the epidemiological inferences made between our previous analysis and our deep sequencing analysis, the sequencing data and bioinformatics pipelines themselves are not directly comparable. Methods to accurately identify hSNPs and incorporate them into transmission analyses are currently an area of active research. We illustrated in our recent paper (*Martin et al., 2018*) that additional steps and strict thresholds must be used to minimize false positive hSNPs, and have conducted the current analysis in consideration of this. However, we note that pre-filtering, our 2015 analysis found that MT-504 had five reference alleles at position 276,685 in the H37Rv alignment (out of 75) and randomly downsampling the current data to simulate ~50 x yielded similar results (5/47 reads at position 276,544 aligned to MT-0080_PB). As most genomic studies of TB employ conservative thresholds of 75–90% allele frequency to classify cSNPs, many bioinformatics pipelines would consider this heterogeneity as potentially suspect at standard sequencing depth. This suggests that greater depth and/or different analytic approaches (e.g., *Wyllie et al., 2019*) are needed to ensure accurate discrimination of sequencing/analytic error from true variation; ultimately, the optimal approach used to identify variants needs to be carefully considered, and appropriate for the study question and data being analyzed.

Finally, while deep sequencing allowed us to detect a novel superspreading event in this context, this approach may not always be necessary; indeed, our previous analysis had identified another super-spreader in the same outbreak using routine sequencing. We acknowledge that this Northern outbreak may not be representative of outbreaks from other settings and/or involving other *M. tuberculosis* lineages. Further studies are needed to quantify the degree to which super-spreading occurs in TB, and examine how and when deep sequencing should be used to detect this.

In summary, we have found evidence of mixed variants with important epidemiologic implications that would not have been detected with standard methods and common filtering criteria. To our knowledge, however, no other studies have been published comparing epidemiological inferences obtained with deep versus routine sequencing for TB outbreak resolution – thus this work represents an important methodological advance in this area. We illustrate that genomic methods, while powerful, still require careful interpretation and can still harbor considerable ambiguity when comparing very close links in a transmission chain, or, as also suggested in *Xu et al. (2019)*, when trying to identify source cases. This finding is likely relevant beyond TB, given the increasing number of pathogens undergoing genomic investigation. Our work also highlights the importance of reproducing previous genomic epidemiology analyses; as the technology and methods used in this field continue to develop, these can lead to improved resolution of transmission and ultimately, challenge previously-held inferences – with critical implications for public health. In terms of TB control, we demonstrate that deep sequencing can aid in transmission analyses, in particular by allowing accurate identification of TB super-spreading events and associated epidemiological characteristics. We propose that deep sequencing is most useful for understanding transmission in settings with low strain diversity, and that these may benefit from routine use of this approach. We hypothesize deep sequencing may also provide additional resolution of transmission events within outbreaks occurring over short, limited timescales – irrespective of local strain diversity, as (by definition) all samples involved in recent transmission would be expected to be closely-related. However, further studies comparing deep versus routine sequencing, ideally from a diversity of clusters and epidemiologic contexts, are needed to fully quantify the added value of this approach for epidemiologic studies of TB.

Overall, this work has important implications for the Canadian North, as well as other regions with high TB transmission; as next-generation sequencing becomes a mainstay in public health surveillance, it is critical we recognize the limitations of analyses done using routine sequencing data. Accurate resolution of transmission is essential for TB control programmes, in order to better understand risk factors for such transmission and enable prioritization of public health resources. With respect to Nunavik, our findings were regarded as very valuable by the regional public health unit and local community leaders; as a direct consequence of this work, ongoing and new investigations of TB genomic epidemiology in Nunavik are using deep sequencing to inform transmission analyses. However, while costs continue to decline, we recognize deep sequencing of all samples in an outbreak may not be economically viable in every setting.

## Acknowledgements

We would like to thank the council of the village for their ongoing support of this work. We also acknowledge staff from the Centre Locales des Services Communautaires and the Nunavik Regional Board of Health and Social Services for their hard work during the outbreak. We would like to thank Dr. Hafid Soualhine (currently at the National Reference Centre for Mycobacteriology in the National Microbiology Laboratory, Public Health Agency of Canada) for previous confirmation of the frameshift deletion in *katG* of MT-6429. We also thank Dr. Anders Gonçalves da Silva and Dr. Glen Carter of the Microbiological Diagnostic Unit Public Health Laboratory at the University of Melbourne for their helpful input on high-quality SMRT sequencing. Library preparation and sequencing for all samples was done at the Genome Québec/McGill Innovation Centre and high-performance computing was done using the Odyssey cluster from the Faculty of Arts and Science, Harvard University. This work was supported by an R01 grant from the National Institutes of Health, awarded to WPH (R01AI128344). RSL was also supported by a Fellowship from the Canadian Institutes of Health Research (MFE 152448) for this work. MAB holds a Canadian Institutes of Health Research Foundation Grant (FDN-148362).

## Additional information

### Funding

| Funder | Grant reference number | Author |
| --- | --- | --- |
| National Institutes of Health | R01AI128344 | William P Hanage |
| Canadian Institutes of Health Research | Fellowship 152448 | Robyn S Lee |
| Canadian Institutes of Health Research | Foundation Award 148362 | Marcel A Behr |

The funders had no role in study design, data collection and interpretation, or the decision to submit the work for publication.

### Author contributions

Robyn S Lee, Conceptualization, Data curation, Software, Formal analysis, Supervision, Funding acquisition, Validation, Investigation, Visualization, Methodology, Project administration; Jean-François Proulx, Data curation, Investigation; Fiona McIntosh, Marcel A Behr, Resources; William P Hanage, Conceptualization, Resources, Supervision, Funding acquisition, Methodology

### Author ORCIDs

Robyn S Lee (iD) https://orcid.org/0000-0001-7120-9053

### Ethics

Human subjects: Ethics approval was obtained from the Institutional Review Board (IRB) of the Harvard T.H. Chan School of Public Health (IRB18-0552) and the IRB of McGill University Faculty of Medicine (IRB A02-M08-18A). Clinical and epidemiological data were previously collected as part of the routine public health response and all data was analyzed in non-nominal fashion, using unique identifiers, therefore individual patient consent was not required. This study was done in collaboration with the Nunavik Regional Board of Health and Social Services.

### Decision letter and Author response

Decision letter https://doi.org/10.7554/eLife.53245.sa1
Author response https://doi.org/10.7554/eLife.53245.sa2

## Additional files

### Supplementary files

• Source data 1. Percent of kmers classified as *Mycobacterium tuberculosis* Complex (MTBC). hSNP frequency is shown using the alignment to MT-0080_PB after removing non-MTBC reads, and filtering with the following thresholds: Phred score < 50, Root Mean Square Mapping Quality [RMS-MQ] $\leq$ 30, depth [DP] < 20, Read Position Rank Sum [ReadPosRankSum] < −8, Fisher Strand Bias [FS] $\geq$ 60

• Source data 2. Comparison of consensus single-nucleotide polymorphisms (cSNPs) and heterogeneous alleles (hSNPs) in all samples aligned to H37Rv versus MT-0080_PB, after initial filtering with Allelic Fraction for cSNPs $\geq$ 0·95 and 0.05 < hSNP < 0·95. Initial filtering thresholds used were: Phred score < **50**, Root Mean Square Mapping Quality [RMS-MQ] $\leq$ **30**, depth [DP] < **20**, Read Position Rank Sum [ReadPosRankSum] < −**8**, Fisher Strand Bias [FS] $\geq$ **60**. As a consequence of the depth of coverage, where allelic fraction was **0·05** < alternative allele [ALT] < **0·95**, all hSNPs had at least 2 REF and ALT alleles by default. This includes all hSNPs and cSNPs identified across all samples, except variants in PE_PGRS and PPE genes, as well as those in mobile elements; some of these variants will be in positions that are excluded from the core alignment, as they failed quality control or are missing in at least one sample in the dataset. Read Position Rank Sum can only be calculated when both reference and alternative alleles are present at a position, therefore the number of cSNPs included in the summary statistics for this variable are 14720 for the H37Rv alignment and 153 for the alignment to MT-0080. *As samples were downsampled to this threshold, this is truncated at 1500.

• Source data 3. Positions with shared heterogeneous alleles (hSNPs) in all 62 samples versus H37Rv.

• Source data 4. Quality metrics for each heterogeneous allele (hSNPs) that was shared across all 62 samples versus H37Rv.

• Source data 5. Assembly metrics for Single Molecule Real-Time sequencing of MT-0080 ('MT-0080_PB'), aligned to NC_000962.3 (H37Rv). Quast (v.5.0.2 *Gurevich et al., 2013*) was used to tabulate the above statistics, with the exception of the number of CDS and RNA, where annotation was done using RASTtk (v.2.0 *Brettin et al., 2015*).

• Source data 6. Core cSNPs and hSNPs between samples, where cSNPs >= 0.95 and 0.05 < hSNP < 0.95 and Phred < 50, RMS-MQ <= 30, DP < 20, FS >= 60, ReadPos < −8, with a minimum of 2 ALT and 2 REF alleles to call hSNPs. This alignment was also filtered to remove variants in PE_PGRS and PE genes, as well as transposons, phage, and integrase as annotated using RASTtk (v.2.0). The original clusters and subgroups identified in *Lee et al. (2015b)* are indicated using different colours. cSNPs and hSNPs are indicated in grey, while cells with alleles that are the same as the reference are filled with white. Due to the minimum DP and allele frequency, a minimum of 2 ALT and 2 REF alleles were needed to call hSNPs by default.

• Source data 7. Core cSNPs and hSNPs between samples, where cSNPs >= 0.95 and 0.05 < hSNP < 0.95 and Phred < 596, RMS-MQ <= 39, DP < 20, FS >= 46, ReadPos < −6, with a minimum of 2 ALT and 2 REF alleles to call hSNPs. This alignment was also filtered to remove variants in PE_PGRS and PE genes, as well as transposons, phage, and integrase as annotated using RASTtk (v.2.0). The original clusters and subgroups identified in *Lee et al. (2015b)* are indicated using different colours. cSNPs and hSNPs are indicated in grey, while cells with alleles that are the same as the reference are filled with white. Due to the minimum DP and allele frequency, a minimum of 2 ALT and 2 REF alleles were needed to call hSNPs by default.

• Source data 8. Core cSNPs and hSNPs between samples, where cSNPs >= 0.99 and 0.01 < hSNP < 0.99 and Phred < 50, RMS-MQ <= 30, DP < 20, FS >= 60, ReadPos < −8, minimum of 2 ALT and 2 REF alleles. This alignment was also filtered to remove variants in PE_PGRS and PE genes, as well as transposons, phage, and integrase as annotated using RASTtk (v.2.0). The original clusters and subgroups identified in *Lee et al. (2015b)* are indicated using different colours. cSNPs and hSNPs are indicated in grey, while cells with alleles that are the same as the reference are filled with white.

• Source data 9. Comparison of consensus single-nucleotide polymorphisms (cSNPs) and heterogeneous alleles (hSNPs) in all samples aligned to H37Rv versus MT-0080_PB, after initial filtering with

Allelic Fraction for cSNPs $\geq$ 0·99 and 0·01 < hSNP < 0·99. Initial filtering thresholds used were: Phred score < **50**, Root Mean Square Mapping Quality [RMS-MQ] $\leq$ **30**, depth [DP] < **20**, Read Position Rank Sum [ReadPosRankSum] < $-$**8**, Fisher Strand Bias [FS] $\geq$ **60**. Where **0·01** < alternative allele [ALT] < **0·99**, a minimum of 2 REF/ALT alleles were required for all hSNPs to reduce risk of including sequencing error; those that failed to meet these criteria were excluded. This includes all hSNPs and cSNPs identified across all samples, except variants in PE_PGRS and PPE genes, as well as those in mobile elements; some of these variants will be in positions that are excluded from the core alignment, as they failed quality control or are missing in at least one sample in the dataset. Read Position Rank Sum can only be calculated when both reference and alternative alleles are present at a position, therefore the number of cSNPs included in the summary statistics for this variable are 13255 for the H37Rv alignment and 148 for the alignment to MT-0080 in the cSNPs $\geq$ **0·99** and **0·01** < ALT < **0·99** analysis. *As samples were downsampled to this threshold, this is truncated at 1500. P values were calculated using on the Wilcoxon-Mann-Whitney test.

- Transparent reporting form

### Data availability

Sequencing data and the assembly for MT-0080 are available on the NCBI's Sequence Read Archive under BioProject PRJNA549270.

The following dataset was generated:

| Author(s) | Year | Dataset title | Dataset URL | Database and Identifier |
|---|---|---|---|---|
| Lee RS, Proulx J-F, McIntosh F, Behr MA, Hanage WP | 2019 | Deep sequencing of a major TB outbreak in the Canadian Arctic | https://www.ncbi.nlm.nih.gov/bioproject/?term=PRJNA549270 | NCBI BioProject, PRJNA549270 |

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
