## [Decision Letter]

**Acceptance summary:**

Tuberculosis infection represents one of the major global health challenges and a better understanding of tuberculosis transmission is critical for designing effective control strategies. This work uses deep sequencing to further investigate a previously studied tuberculosis transmission cluster in Canada. It uses deep sequencing to identify co-infection of a single individual with two closely related strains, and suggests that both strains were subsequently transmitted. This increases the number of individuals thought to have been infected from a single source individual, and thus increases our estimates of the potential frequency and magnitude of 'super-spreading' events. Although this represents analysis of a single infection cluster, such detailed studies also provide a model for future outbreak investigation.

**Decision letter after peer review:**

Thank you for submitting your article "Previously undetected super-spreading of *Mycobacterium tuberculosis* revealed by deep sequencing" for consideration by *eLife*. Your article has been reviewed by three peer reviewers, and the evaluation has been overseen by Miles Davenport as Reviewing Editor, and Eduardo Franco as the Senior Editor. The following individual involved in review of your submission has agreed to reveal their identity: Conor J Meehan.

As is customary in *eLife*, the reviewers have discussed the reviews with one another and the Reviewing Editor has drafted this decision to help you prepare a revised submission.

Summary:

Lee et al. report a re-analysis of a tuberculosis outbreak combining a local reference derived from long-reads and ultra-deep sequencing to reveal minor variants. They use the novel information to identify a previously undetected super-spreader. This is very carefully done piece of work into which the authors have clearly put a lot of thought. Although mixed bacterial populations have been used to identify index cases and so-called super-spreaders from WGS TB data in the past, the authors have given this issue greater focus here and emphasised the point that such mixed populations might not be detected without sequencing to a greater depth. This work is an excellent example of the benefits of this deep sequencing and outlines both the need for local outbreak reference sequences and associated hSNP analysis in low diversity settings.

Essential revisions:

1) The crux of this work is about identifying MT-504 as a super spreader. From the core SNP tree in Figure 1A this would not be evident at all based on their location in a subgroup of the sublineage and not in a position closer to the root. The use of hSNPs to uncover this is well done but the paragraph is difficult to read. Indeed only because this is my field do I think I was able to follow. Perhaps authors could construct a phylogenetic network of this subsection to show how the MT-504 resides as a central hub compared to the tree? In any case this section needs to be made easy to read as without it the more un-accustomed reader will get lost.

2) This work is most important for low diversity settings. This should be better highlighted throughout, perhaps with a comment on how this could be applicable to high diversity settings, if at all. Many Mtb researchers think all new WGS work is applicable in all settings and it would be of help if the authors demonstrated where and when people need to make the extra effort and indeed massively extra cost of deep sequencing.

3) The portion on super spreaders being found at diagnosis in the Discussion is interesting but not directly related to the findings here. We are a long way from identifying super spreaders based on some genomic trait (especially as this could be host related) and the WGS work here won't lead to such clinical decision making. I suggest this section be vastly shortened or removed so readers don't think this is a possibility from WGS.

4) A major criticism is that although this is a carefully crafted analysis, it remains essentially anecdotal. To draw conclusions that deep sequencing is now ideally required for all outbreaks is probably a little overenthusiastic. A systematic analysis of far larger number of clusters is required to assess the importance of mixed populations that go undetected at routine sequencing depth before conclusions can be drawn about what should and shouldn't be routine practice. It could easily be that most mixed populations that are helpful to inferences around directionality are detectable at routine sequencing depth in the vast majority of outbreaks. That needs to be established. Thus, although the discussion assumes that superspreaders can be discovered following the approach described, it is not totally clear this is always the case. This manuscript only offers one example, so this should be mentioned as a limitation but also as a call for future work to find out how common is the phenomenon and whether it can be revealed by ultra-deep sequencing.

5) The same criticism could be levelled at the conclusion that a local reference is important. It was in this case, but data has not been presented from which to conclude that it is routine necessity.

---

## [Author Response]

Essential revisions:1) The crux of this work is about identifying MT-504 as a super spreader. From the core SNP tree in Figure 1A this would not be evident at all based on their location in a subgroup of the sublineage and not in a position closer to the root. The use of hSNPs to uncover this is well done but the paragraph is difficult to read. Indeed only because this is my field do I think I was able to follow. Perhaps authors could construct a phylogenetic network of this subsection to show how the MT-504 resides as a central hub compared to the tree? In any case this section needs to be made easy to read as without it the more un-accustomed reader will get lost.

We apologize for any lack of clarity. It has been a challenge to present a comparison with previous work, wherein we need to show our results clearly but also be mindful of not replicating previously published text or figures.

The consensus SNP tree in Figure 1A (Figure 2A in the revised version, as the supplementary figure was moved to the main text) included all available samples from this village from 2007-2012 (i.e., 62/65 outbreak and historical cases, not only from IIIB, but IIIA, IIIC and VA as well). We did this to illustrate that, despite changes in sequencing platforms and analysis, we still recapitulate the original three clusters (IIIA/B/C) from this community. We have added the following clarification to the Results: “All 62 samples (historical and outbreak) were included, for comparison with our previous work.” Based on our previous work, cluster IIIA (which was first seen in the community in 2007, while IIIB and IIIC were not seen until 2009 and 2012, respectively) was ancestral to clusters Mj-IIIB and IIIC. We would therefore not expect MT-504, which is in cluster IIIB, to be at the root of this particular tree.

In response to the reviewers’ comment, we have done the following to help make this figure and corresponding section of the text more clear for *eLife* readers:

A) We have extensively revised the section entitled “hSNPs identify super-spreaders and more accurately resolve transmission clusters” to walk the reader through the data in more detail (hSNPs identify super-spreaders and more accurately resolve transmission clusters).

B) We have added the informative alleles to this figure (2A in the revised version)

C) We have also added a *new* epidemiologic curve showing the cases stratified by sub-group of transmission according to the original analysis to Figure 1B (2B in the revised version), to assist the reader in understanding the timeline.

On reviewing the raw fasta file, we also wanted to report that we realized some missing sites had inadvertently been included despite using a well-known tool (‘snp-sites’) to exclude all non-AGTC characters for this step. This led to minor differences in the maximum likelihood tree structure from the submitted version. This has now been updated (and did not alter our overall epidemiologic inferences).

2) This work is most important for low diversity settings. This should be better highlighted throughout, perhaps with a comment on how this could be applicable to high diversity settings, if at all. Many Mtb researchers think all new WGS work is applicable in all settings and it would be of help if the authors demonstrated where and when people need to make the extra effort and indeed massively extra cost of deep sequencing.

We thank the reviewers for this comment. It is true that the current study examines the application of deep sequencing to an outbreak in a low-diversity setting. However, given the low mutation rate of *M. tuberculosis*, samples from cases involved in recent transmission (e.g., within 2 years), can often have 0 consensus SNPs between them. Indeed, our collaborators at the US Center for Disease Control’s Division of TB Elimination have reported to us difficulties resolving recent transmission using consensus SNPs alone for this reason; we therefore anticipate deep sequencing could help provide additional resolution, by helping resolve person-to-person transmission events within these groups of transmission. In this case, the utility of deep sequencing in this case would be irrespective of the diversity of strains circulating in the setting. We do, however, agree that deep sequencing would likely be more useful in low-diversity settings for surveillance over longer time scales than it would be in high-diversity settings, as more fixed variation would be expected to occur over samples spread over greater time periods.

We had previously stated that we believe deep sequencing may be useful for “investigation of outbreaks occurring over short time scales and/or in settings such as the Canadian North, where the genetic diversity of circulating strains is especially low. In both of these circumstances, it is common to find many samples separated by zero cSNPs, hindering accurate source ascertainment.” We have also added the following statement to the Discussion: “We propose that deep sequencing is most useful for understanding transmission in settings with low strain diversity, and that these may benefit from routine use of this approach. […] However, further studies comparing deep versus routine sequencing, ideally from a diversity of clusters and epidemiologic contexts, are needed to fully quantify the added value of this approach for epidemiologic studies of TB.” We also discuss representativeness of this dataset further under point #4 below.

3) The portion on super spreaders being found at diagnosis in the Discussion is interesting but not directly related to the findings here. We are a long way from identifying super spreaders based on some genomic trait (especially as this could be host related) and the WGS work here won't lead to such clinical decision making. I suggest this section be vastly shortened or removed so readers don't think this is a possibility from WGS.

Thank you for your feedback on this section. We have cut this text from the manuscript to avoid further confusion, however, to clarify, we did not mean to imply that deep sequencing could be used to identify *bacterial* characteristics predictive of future super-spreading events. We absolutely agree with the reviewer that such a goal is generally ‘a long way off’, and, with respect to Nunavik, our previous work (Lee et al., 2015) actually suggested that bacterial characteristics are not likely to be driving transmission in this region. Furthermore, even if there were such bacterial characteristics, delays in sequencing (due to shipping samples south, batching of samples, need for culture, etc.) generally mean these findings would not be available in real-time to assist with outbreak response. What we are instead proposing to use deep sequencing to help accurately discriminate super-spreading events, and subsequently, identify epidemiological and clinical characteristics that are associated with these using more ‘classical’ epidemiologic methods (i.e., regression analyses). Dr. Lee is currently leading such a study for this region. These communities have very limited resources in terms of clinical staff, who are typically managing numerous routine health needs in addition to TB; if we can help identify the most likely clinical and epidemiological characteristics of these historical super-spreaders, nursing staff and physicians can potentially triage cases with these characteristics in the future as they are diagnosed, and better prioritize investigation of their contacts.

4) A major criticism is that this is a carefully crafted analysis, it remains essentially anecdotal. To draw conclusions that deep sequencing is now ideally required for all outbreaks is probably a little overenthusiastic. A systematic analysis of far larger number of clusters is required to assess the importance of mixed populations that go undetected at routine sequencing depth before conclusions can be drawn about what should and shouldn't be routine practice. It could easily be that most mixed populations that are helpful to inferences around directionality are detectable at routine sequencing depth in the vast majority of outbreaks. That needs to be established. Thus, although the discussion assumes that superspreaders can be discovered following the approach described, it is not totally clear this is always the case. This manuscript only offers one example, so this should be mentioned as a limitation but also as a call for future work to find out how common is the phenomenon and whether it can be revealed by ultra-deep sequencing.

We appreciate the reviewers’ feedback, and we agree that the present work is not sufficient to offer guidelines for the study of all outbreaks, in settings with high and low strain diversity and TB burden. However, while this is indeed one outbreak, it is an essential and necessary step to enable such work in the future. We had in this study a unique opportunity to compare approaches for inferring transmission under essentially ideal conditions, i.e., where (due to the scale of the outbreak, and geography of Nunavik, limiting out-migration), there was all cases were detected, and detailed epidemiological data on each was available. In departing from the standard approaches to genomic epidemiology, this paper is a crucial starting point for future work; further studies are needed – with deep and regular sequencing data from all samples, along with epidemiological data – in order to assess the added value of deep sequencing across settings and contexts. In line with point #2 above, we have modified the Discussion to address this.

In addition to this, we have also added the following: “Finally, while deep sequencing allowed us to detect a novel superspreading event in this context, this approach may not always be necessary; indeed, our previous analysis had identified another super-spreader in the same outbreak using routine sequencing. […] Further studies are needed to quantify the degree to which super-spreading occurs in TB, and examine how and when deep sequencing be used to detect this.” We have also removed our suggestion in the Discussion that deep sequencing may be needed for all smear-positive cases, to make the text less prescriptive.

5) The same criticism could be levelled at the conclusion that a local reference is important. It was in this case, but data has not been presented from which to conclude that it is routine necessity.

We found a significant reduction in what appear to be false positive hSNPs using a local reference genome, and do think such a reference is useful to help minimize these. We have been careful to indicate that this may not be necessary for cSNP detection for TB (though we would note, in our experience, that a local reference is often preferred for this purpose for other pathogens). Our findings are also consistent with other studies that have looked at virulence factors; the H37Rv strain has been passaged numerous times since its isolation, and gene content is not identical to clinical strains. However, in response to the reviewer’s comment, we have modified our text from “We propose these [i.e., local reference genomes] are also warranted for hSNP analysis” to “Our analysis *suggests* that these *may* also be warranted for hSNP analysis”. We have also changed our ‘recommendation’ to a ‘suggestion’ that “where it is possible to do so – it would be best to generate and use a local reference”.